# Impact of integrating objective structured clinical examination into academic student assessment: Large-scale experience in a French medical school

Alexandre Matet [1,2,3☯]*, Ludovic Fournel[1,4,5☯], François Gaillard[6☯], Laurence Amar[1,7,8], Jean-Benoit Arlet[1,9], Stéphanie Baron[1,6], Anne-Sophie Bats[1,10,11], Celine Buffel du Vaure[12], Caroline Charlier[1,13,14], Victoire De Lastours[1,15,16], Albert Faye[1,17], Eve Jablon[18], Natacha Kadlub[1,19], Julien Leguen[1,20], David Lebeaux[1,21], Alexandre Malmartel [12], Tristan Mirault[1,7,8], Benjamin Planquette[1,22,23], Alexis Régent[1,24,25], Jean-Laurent Thebault[12], Alexy Tran Dinh[1,26,27], Alexandre Nuzzo [1,28], Guillaume Turc[1,29,30], Gérard Friedlander[1,6,31], Philippe Ruszniewski[1,29,32], Cécile Badoual[1,7,33], Brigitte Ranque[1,7,9], Mehdi Oualha[1,34,35‡], Marie Courbebaisse[1,6,31‡]

1 Université de Paris, Faculté de Médecine, Paris, France, 2 Centre de Recherche des Cordeliers, INSERM UMR1138, Paris, France, 3 Service d'ophtalmologie, Institut Curie, Paris, France, 4 INSERM UMR1124, Paris, France, 5 Service de chirurgie thoracique, AP-HP, Hôpital Cochin, Paris, France, 6 Département de physiologie, AP-HP, Hôpital Européen Georges Pompidou, Paris, France, 7 PARCC INSERM U970, Paris, France, 8 Département d'hypertension artérielle, AP-HP, Hôpital Européen Georges Pompidou, Paris, France, 9 Service de Médecine interne, AP-HP, Hôpital Européen Georges Pompidou, Paris, France, 10 INSERM UMR-S 1147, Paris, France, 11 Service de gynécologie oncologique et de chirurgie du sein, AP-HP, Hôpital Européen Georges Pompidou, Paris, France, 12 Département de médecine générale, Université de Paris, Faculté de Médecine, Paris, France, 13 Institut Pasteur, INSERM U1117, Paris, France, 14 Département de maladies infectieuses et tropicales, AP-HP, Hôpital Universitaire Necker, Paris, France, 15 IAME, UMR1137, INSERM, Paris, France, 16 Service de Médecine Interne, AP-HP, Hôpital Beaujon, Clichy, France, 17 Service de Pédiatrie Générale, Hôpital Robert Debré, Paris, INSERM ECEVE 1123, Paris, France, 18 Service AGIR, Université de Paris, Faculté de Médecine, Paris, France, 19 Département de chirurgie maxillo-faciale et de chirurgie plastique, AP-HP, Hôpital Universitaire Necker, Paris, France, 20 Service de Gériatrie, AP-HP, Hôpital Européen Georges Pompidou, Paris, France, 21 Département de Microbiologie, AP-HP, Hôpital Européen Georges Pompidou, Paris, France, 22 INSERM UMR S1140, Paris, France, 23 Service de Pneumologie et de soins intensifs, AH-HP, Hôpital Européen Georges Pompidou, Paris, France, 24 Institut Cochin, INSERM U1016, CNRS UMR 8104, LabEx INFLAMEX, Paris, France, 25 Service de Médecine Interne, Centre de Référence pour les Maladies Systémiques Auto immunes Rares d'Ile-de-France, AP-HP, Hôpital Cochin, Paris, France, 26 INSERM U1148 LVTS, Villetanneuse, France, 27 Département d'Anesthésie-Réanimation, AP-HP, Hôpital Bichat-Claude Bernard, Paris, France, 28 Service de gastro-entérologie et pancréatologie, AP-HP, Hôpital Beaujon, Paris, France, 29 INSERM U1266, Paris, France, 30 Service de neurologie, Hôpital Sainte Anne, AH-HP, Paris, France, 31 Institut Necker-Enfants Malades, INSERM U1151-CNRS UMR8253, Paris, France, 32 INSERM UMR1149, Paris, France, 33 Service d'anatomopathologie, AP-HP, Hôpital Européen Georges Pompidou, Paris, France, 34 Unité EA7323, Université de Paris, Faculté de Médecine, Paris, France, 35 Service de réanimation et de surveillance continue médico-chirurgicale pédiatrique, AP-HP, Hôpital Universitaire Necker, Paris, France

☯ These authors contributed equally to this work.
‡ These authors also contributed equally to this work.
* alexandre.matet@curie.fr



**Data Availability Statement:** The S1 Table file is available from the Figshare repository : https://doi.org/10.6084/m9.figshare.13507224.v1.

## Abstract

### Purpose

Objective structured clinical examinations (OSCE) evaluate clinical reasoning, communication skills, and interpersonal behavior during medical education. In France, clinical training

**Funding:** The authors received no specific funding for this work.

**Competing interests:** The authors have declared that no competing interests exist.

has long relied on bedside clinical practice in academic hospitals. The need for a simulated teaching environment has recently emerged, due to the increasing number of students admitted to medical schools, and the necessity of objectively evaluating practical skills. This study aimed at investigating the relationships between OSCE grades and current evaluation modalities.

## Methods

Three-hundred seventy-nine 4th-year students of University-of-Paris Medical School participated to the first large-scale OSCE at this institution, consisting in three OSCE stations (OSCE#1–3). OSCE#1 and #2 focused on cardiovascular clinical skills and competence, whereas OSCE#3 focused on relational skills while providing explanations before planned cholecystectomy. We investigated correlations of OSCE grades with multiple choice (MCQ)-based written examinations and evaluations of clinical skills and behavior (during hospital traineeships); OSCE grade distribution; and the impact of integrating OSCE grades into the current evaluation in terms of student ranking.

## Results

The competence-oriented OSCE#1 and OSCE#2 grades correlated only with MCQ grades ($r = 0.19$, $P<0.001$) or traineeship skill grades ($r = 0.17$, $P = 0.001$), respectively, and not with traineeship behavior grades ($P>0.75$). Conversely, the behavior-oriented OSCE#3 grades correlated with traineeship skill and behavior grades ($r = 0.19$, $P<0.001$, and $r = 0.12$, $P = 0.032$), but not with MCQ grades ($P = 0.09$). The dispersion of OSCE grades was wider than for MCQ examinations ($P<0.001$). When OSCE grades were integrated to the final fourth-year grade with an incremental 10%, 20% or 40% coefficient, an increasing proportion of the 379 students had a ranking variation by ±50 ranks ($P<0.001$). This ranking change mainly affected students among the mid-50% of ranking.

## Conclusion

This large-scale French experience showed that OSCE designed to assess a combination of clinical competence and behavioral skills, increases the discriminatory capacity of current evaluations modalities in French medical schools.

## Introduction

Objective structured clinical examination (OSCE) aims to evaluate performance and skills of medical students including clinical reasoning, communication skills, and interpersonal behavior [1–4]. OSCE has been proposed as a gold standard for the assessment of medical students performance during the 'clinical' years of medical school [5, 6] and is used in several countries worldwide [7–10], including the United States and Canada [11–13] who pioneered its integration in medical teaching programs.

The use of OSCE is currently expanding in France, where clinical training has long relied on bedside clinical practice in academic hospitals. To date, in this country, medical knowledge is mainly evaluated using multiple choice questions (MCQ)-based written examinations, whereas the evaluation of clinical skills and behavior relies on subjective assessments at the

end of each hospital-based traineeship in a non-standardized manner. Upon completion of the sixth year of medical school, all French students take a final classifying national exam that determines their admission into a residency program. Their admission into a given specialty and a given teaching hospital network is based on their national rank. This national exam is currently based on MCQs only, either isolated or related to progressive clinical cases, and MCQs dealing with the critical reading of a peer-reviewed medical article.

However, the need for a simulated teaching environment has recently emerged in French medical schools, due to the increasing number of admitted students, and the necessity of objectively evaluating practical skills. In a near future, OSCE will be implemented in the reformed version of the French final classifying national exam, accounting for 40% of the final grade. In this context, medical teachers at the Université de Paris Medical School Paris, France, which has two sites that have recently merged, the Paris Nord and Paris Centre sites, and ranks among the largest medical schools in France with 400–450 students per study year, have designed a large-scale OSCE taken by all fourth-year medical students to assess the impact of such evaluation on student ranking.

Considering the plurality of evaluation modalities available for medical students, to study the correlations between grades obtained on performance-based tests, such as OSCE, and other academic and non-academic tests, is of paramount importance. The aims of this study were (i) to investigate the correlation of OSCE grades with those obtained on current academic evaluation modalities, consisting in written MCQ-based tests and assessment of clinical skills and behavior during hospital traineeship, (ii) to analyze the distribution of grades obtained on this first large-scale experience of OSCE at this institution, and (iii) to simulate the potential impact of integrating OSCE grades into the current evaluation system in terms of student ranking.

## Methods

### Study population

The 426 medical students completing the fourth year at the Paris Centre site of Université de Paris Medical School (Paris, France), from September 2018 to July 2019 were invited to participate to the large-scale OSCE evaluation organized by the Medical School on May 25, 2019. Students were exempted of OSCE if they were on night shift the night before, or the day of the OSCE, or if they were completing a traineeship abroad at the time of the evaluation (European student exchange program). The education council and review board of University of Paris approved the observational and retrospective analysis of grades obtained at OSCE and all written and practical evaluations during the 2018–2019 academic year for the fourth-year class. The need for informed consent was waived because all data were anonymized before analysis.

### Current evaluation of fourth-year medical students

**Hospital-based traineeship evaluation.** At the end of each 3-month hospital traineeship, students are evaluated by the supervising MDs in a non-standardized manner in two areas: i) knowledge and clinical skills acquired during the traineeship (50% of the grade of the traineeship) and ii) behavior, which includes presence, diligence, relationship to the patient, integration within the care team (50% of the grade of the traineeship).

**Academic evaluation.** During the fourth year of medical school, students are divided in three subgroups and enrolled successively in three teaching units (TU) subdivided as follows: TU1 includes cardiology, pneumology, and intensive care; TU2 includes hepato-gastroenterology, endocrinology, and diabetology; and TU3 includes rheumatology, orthopedics, and dermatology. For each subgroup, the evaluation of each TU takes place at the end of the quarter

during which the three specialties of this TU were taught. Thus, the whole class is not evaluated concomitantly for a given TU.

The academic evaluation of each TU lasts 210 minutes. This test comprises three progressive clinical cases including 10 to 15 MCQs, 45 isolated MCQs (15 MCQs per specialty taught in the unit), and 15 MCQs evaluating the critical reading of a scientific article related to one of the specialties taught in the TU.

**Calculation of the final grade for each unit of teaching.** The academic evaluation accounts for 90% of the final grade for a TU, and the grade obtained from the evaluation of knowledges and medical skills obtained at the end of the hospital-based traineeship corresponding to this TU accounts for 10%. The grade obtained from the evaluation of behavior during the hospital traineeship is used to pass the traineeship but is not taken into account in the TU average grade. To pass a given TU, a minimal grade of 50% ($\geq$10/20) must be obtained.

**OSCE stations.** OSCE scenarios were designed by a committee of 16 medical teachers, according to the guidelines of the Association for Medical Education in Europe [14, 15]. The first OSCE station (OSCE #1) focused on diagnosis (acute dyspnea due to pulmonary embolism secondary to lower leg deep venous thrombosis), the second (OSCE #2) on prevention (cardiovascular counselling after acute myocardial infarction) and the third (OSCE #3) on relational skills (exposition of cholecystectomy indication following acute cholecystitis). The OSCE #1, #2 and #3 scenarios and their detailed standardized evaluation grids are presented in **S1–S3 Data**, respectively. Of note, the first and second stations (OSCE #1 and OSCE #2) dealt with cardiovascular conditions covered in TU1, whereas the third OSCE station (OSCE #3) was a hepato-gastroenterology scenario and therefore corresponded to TU2. The items retained in the evaluation grid to assess student's performance followed the guidelines of the Association for Medical Education in Europe, which outlines four major categories: clinical cognitive and psychomotor abilities (grouped and referred to as 'Competence'); non-clinical skills and attitudes (grouped and referred to as 'Behavior') [14]. This categorization of items revealed that OSCE #1, OSCE#2 and OSCE #3 were designed to assess clinical competence and relational skills in difference proportions, as displayed in **Fig 1**.

Physicians and teachers from all clinical departments at Université de Paris Medical School were enrolled as actors to act as standardized patients. The OSCE committee organized several training sessions to explain the script of each OSCE station and ensure standardization of actions and dialogues from standardized patients. Moreover, each OSCE scenario was

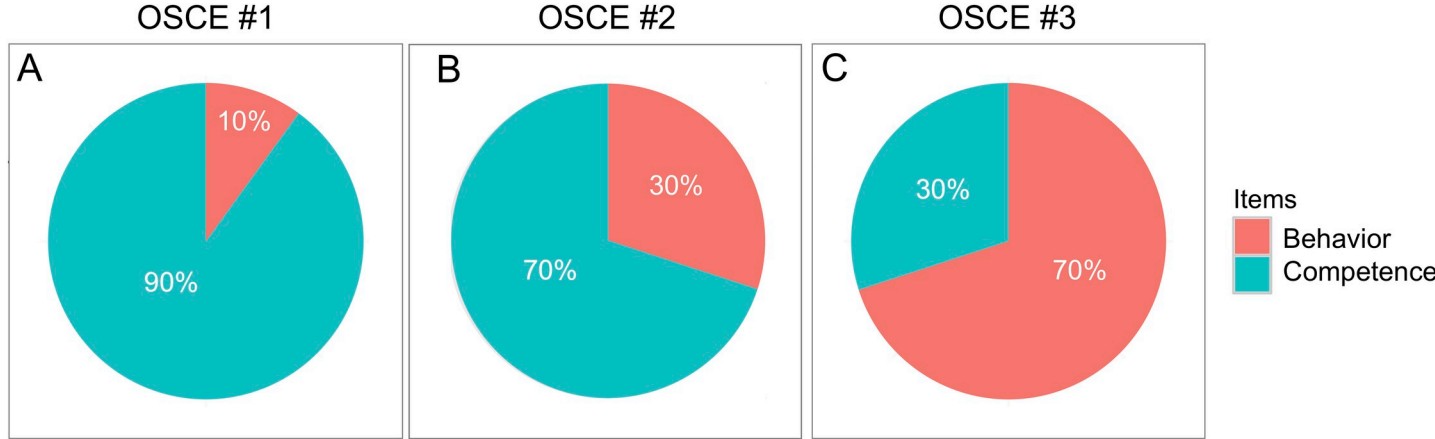

**Fig 1.** Pie charts displaying the proportions of competence-based and behavior-based items in the evaluation grids for OSCE stations #1, #2 and #3 (A, B and C, respectively). Detailed evaluation grids are provided as S1–S3 Data.

recorded by members of the OSCE committee who had contributed to the scripts, and the videos were available in a secure online platform for training.

**Organization of the OSCE.**   The test took place on May, 2019 concomitantly for all participating students, at three different facilities of Université de Paris, Paris Centre site (Cochin, Necker, and European Georges-Pompidou University Hospitals, Paris, France). The duration of each station was 7 minutes. In each room, two teachers were present: one acted as standardized patient, and the second evaluated the performance of each student in real time according to a standardized evaluation grid (provided with the OSCE scripts in **S1**–**S3 Data**), which was accessed on a tablet connected to the internet. In addition to the 16 members of the OSCE organization committee, 162 teachers of Université de Paris participated as standardized patients or evaluators. To assess quality and inter-standardized patient reproducibility, OSCE coordinators attended as observers at least one OSCE scenario run by each standardized patient. The homogeneity of training between assessors was maximized by preparatory meetings throughout the academic year preceding the OSCE test, specific training for each OSCE station in small groups by one single coordinating team, diffusion of video recordings of a standard patient undergoing each OSCE station. Moreover, the homogeneity of motivation between assessors was maximized by the facts that all were medical doctors belonging to the same university hospital network, that all were implicated at various degrees in medical pedagogy, and that all participated for the first time to a large-scale pedagogical experiment of an upcoming evaluation and teaching modality.

The proportion of evaluators from the same specialty as the one evaluated in each OSCE station (pneumologists in OSCE1, cardiologists/vascular specialists in OSCE2, and gastroenterologists/digestive surgeons in OSCE3) was ~9.5% across the 3 stations. This proportion was 7%, 12%, and 9% for OSCE 1, 2 and 3, respectively.

## Statistical analyses

Descriptive and correlative statistics were computed on GraphPad Prism (version 5.0f, GraphPad Software). Spearman correlation coefficients, and Mann-Whitney tests were used where appropriate, due to the non-normal distribution of grades (ascertained by the density plot as shown in **Fig 2** and confirmed by the Kolmogorov-Smirnov test, $P<0.001$ for the distribution of OSCE, MCQ, traineeship skill and traineeship behavior grades). Categorical distributions were compared using the Chi-square test. Plots were created using the R Software (Version 3.3.0, R Foundation for Statistical Computing, R Core Team, 2016, http://www.R-project.org/) and the 'ggplot' package. Multivariate analyses were conducted using R software.

For certain analyses, competence-oriented items and behavior-oriented items were extracted from OSCEs #1, #2, and #3 and averaged, as previously reported by Smith et al. [16].

To compare the score obtained from OSCE to the current evaluation based on MCQ tests, we simulated the potential impact of the integration of OSCE scores on the ranking of fourth-year medical students included in our study. Since the evaluation of teaching units and the national classifying exam both consist in MCQs (isolated or based on progressive medical cases or on critical reading of a peer-reviewed medical article), we first classified fourth-year medical students according to the mean grades obtained in the three teaching units, as if it was the classification they would have obtained on the national classifying exam. To evaluate the potential impact of OSCE on the rankings, we integrated the mean grade for the three OSCE stations into the current evaluation, with 10%, 20% and 40% coefficients (based on the planned coefficient of 40% for OSCE in the future version of the final classifying exam). We evaluated the proportion of students who would enter the top 20% and who would be dropped, upon OSCE grade inclusion with 20% or 40% coefficients.

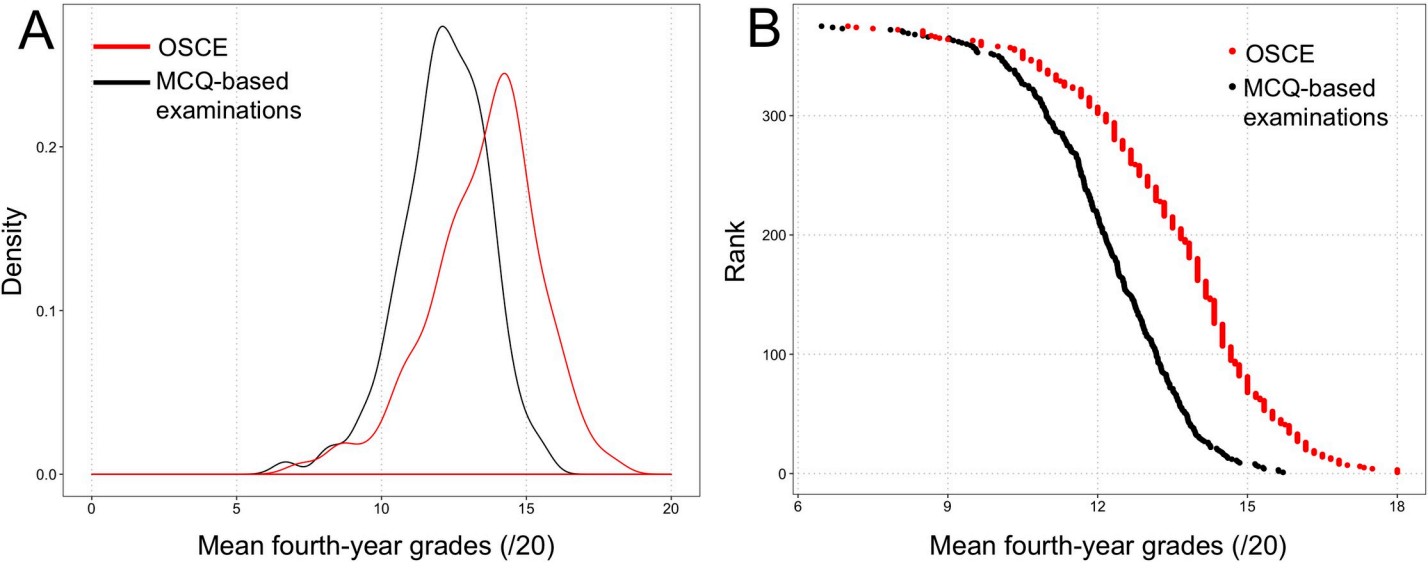

**Fig 2. Distribution of mean OSCE grades (red) and mean fourth-year multiple-choice question (MCQ)-based grades (black).** (A) Density plot showing the wider dispersion of OSCE grades compared to MCQ grades. (B) Relationship between student rank among the 379-student class, and grades obtained at OSCE and MCQ-based examinations, showing a flatter slope for OSCE and a steeper slope for MCQs, confirming the wider dispersion of OSCE grades than MCQ grades among the fourth-year class.

## Results

Of 426 students completing the fourth year at Université de Paris Medical School, Paris Centre site, from September 2018 to July 2019, 379 (89%) participated in the first large-scale OSCE test. The descriptive statistics of the average fourth-year MCQ-based grades obtained after the three TU, each OSCE station, and the mean OSCE grades, are summarized in **Table 1**. Grades obtained at each OSCE station are provided in the S1 Table (https://doi.org/10.6084/m9.figshare.13507224.v1).

### Correlation between OSCE, MCQ-based grades and hospital-based traineeship grades

Correlations between OSCE grades and MCQ-based grades obtained for each TU, traineeship skills, and traineeship behavior are explored in **Fig 3** and **Table 2**. Positive, but weak correlations were identified between the mean OSCE grade and the mean fourth-year MCQ-based examination or traineeship skill grades ($r = 0.18$, $P = 0.001$ and $r = 0.19$, $P<0.001$, **Fig 3A, 3B**, respectively). Interestingly, mean OSCE grades did not correlate with traineeship behavior

**Table 1. Descriptive statistics for mean multiple-choice-based examination grades and OSCE grades of the fourth-year class of medical school.**

|  | MCQ-based examinations | Mean OSCE | OSCE #1 | OSCE #2 | OSCE #3 |
|---|---|---|---|---|---|
| Mean | 12.17 | 13.53 | 12.66 | 13.33 | 14.55 |
| SD | 1.51 | 1.92 | 3.11 | 2.74 | 2.75 |
| Median | 12.25 | 13.83 | 13.0 | 13.50 | 15.0 |
| Range | 6.47–15.71 | 7.0–18.0 | 2.0–19.0 | 3.0–19.0 | 4.0–20.0 |

In the French notation system, the maximal grade is 20.

SD = standard deviation; MCQ = multiple-choice question; OSCE = objective structured clinical examination

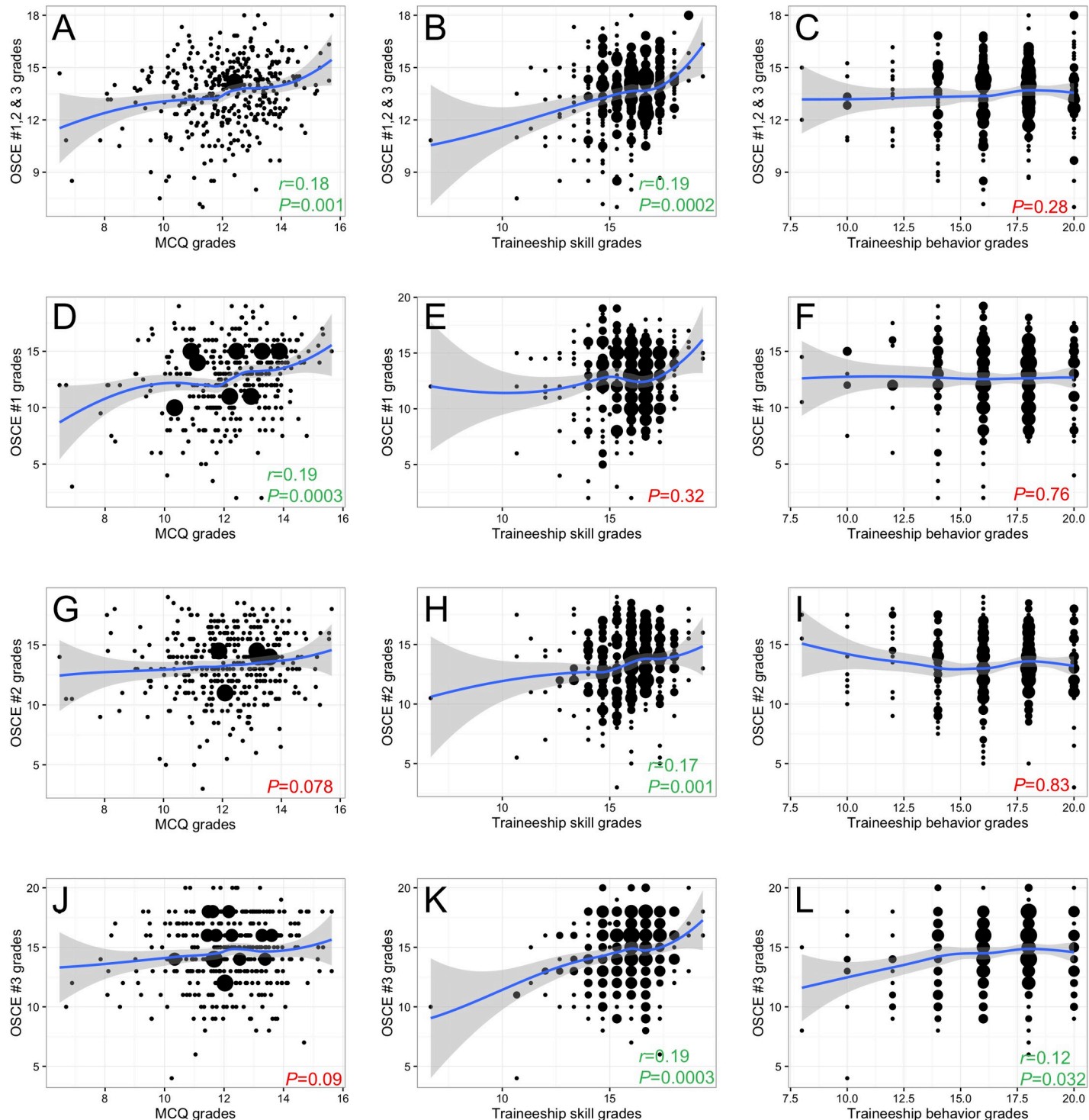

**Fig 3. Scatterplots of the relationships between OSCE grades and mean fourth-year teaching unit grades.** (A-C) Mean OSCE grades versus mean fourth-year multiple-choice question (MCQ)-based examination grades, traineeship skill and traineeship behavior grades. (D-F) Mean OSCE #1 grades versus mean fourth-year MCQ-based examination, traineeship skill and traineeship behavior grades. (G-I) Mean OSCE #2 grades versus mean fourth-year MCQ-based examination, traineeship skill and traineeship behavior grades. (J-L) Mean OSCE #3 grades versus mean fourth-year MCQ-based examination, traineeship skill and traineeship behavior grades. To highlight trends, a smoothing regression line was added to each plot using the geom_smooth function (R Software, ggplot2 package). *P* values and Spearman *r* coefficient were highlighted in green for significant and in red for non-significant correlations, respectively.

Table 2. Correlation between OSCE grades and mean fourth-year multiple-choice-question-based grades.

| | Spearman *r* | *P* |
|---|---|---|
| **Mean 4th-year MCQ-based examination grades** | | |
| vs. OSCE mean | 0.18 | 0.001 |
| vs. OSCE #1 | 0.19 | 0.0003 |
| vs. OSCE #2 | - | 0.078 |
| vs. OSCE #3 | - | 0.094 |
| **Mean 4th-year traineeship skills grade** | | |
| vs. OSCE mean | 0.19 | 0.0002 |
| vs. OSCE #1 | - | 0.32 |
| vs. OSCE #2 | 0.17 | 0.001 |
| vs. OSCE #3 | 0.19 | 0.0003 |
| **Mean 4th-year traineeship behavior grade** | | |
| vs. OSCE mean | - | 0.28 |
| vs. OSCE #1 | - | 0.76 |
| vs. OSCE #2 | - | 0.83 |
| vs. OSCE #3 | 0.12 | 0.032 |

OSCE = objective structured clinical examination; MCQ = multiple-choice question.

OSCEs #1 and #2 are focused on cardiovascular and are predominantly competence-oriented; OSCE #3 is focused on hepato-gastroenterology, and predominantly behavior-oriented.

grades (*P* = 0.28, **Fig 3C**). A sub-analysis revealed that grades obtained at each OSCE station correlated differently with the other evaluation modalities. OSCE #1 grades correlated with MCQ-based grades (*r* = 0.19, *P*<0.001, **Fig 3D**), but not traineeship skill or behavior grades (*P* = 0.32 and *P* = 0.76, **Fig 3E and 3F**, respectively). OSCE#2 grades showed a near-significant correlation with MCQ-based grades (*P* = 0.078, **Fig 3G**), a correlation with traineeship skill grades (*r* = 0.17, *P* = 0.001, **Fig 3H**) but not with traineeship behavior grades (*P* = 0.83, **Fig 3I**). Conversely, OSCE #3 grades correlated with both traineeship skill and behavior grades (*r* = 0.19, *P*<0.001 and *r* = 0.12, *P* = 0.032, **Fig 3K and 3L**, respectively), but not with MCQ-based grades (*P* = 0.09, **Fig 3J**).

Moreover, of 94 students within the top quarter of the fourth-year class (top 25%) for averaged MCQ-based grades, only 27 students (29%) obtained an averaged OSCE grade (average of OSCE #1–3) within the top quarter. In contrast, 39 (41%) of the 94 students within the top quarter for traineeship skill grades, and 55 (59%) of the 94 students within the top quarter for traineeship behavior grades obtained an averaged OSCE grade within the top quarter (*P*<0.001, Chi-square test, **Fig 4**).

**Table 3** summarizes an additional analysis averaging separately all competence-oriented and all behavior-oriented items from the three OSCE stations. Whereas averaged behavior-oriented items showed a significant correlation to traineeship skill and behavior grades but not to MCQ-based grades (*r* = 0.13, *P* = 0.010; *r* = 0.11, *P* = 0.046, and *P* = 0.079, respectively), averaged competence-oriented items showed a significant correlation to MCQ-based and traineeship skill grades, but not to traineeship behavior grades (*r* = 0.15, *P* = 0.004; *r* = 0.15, *P* = 0.003 and *P* = 0.35, respectively).

## Distribution of grades obtained on the OSCE

As shown in **Table 1**, grades obtained at the behavior-oriented OSCE #3 station were higher than those obtained at the predominantly competence-oriented OSCE #1 and #2 stations. The

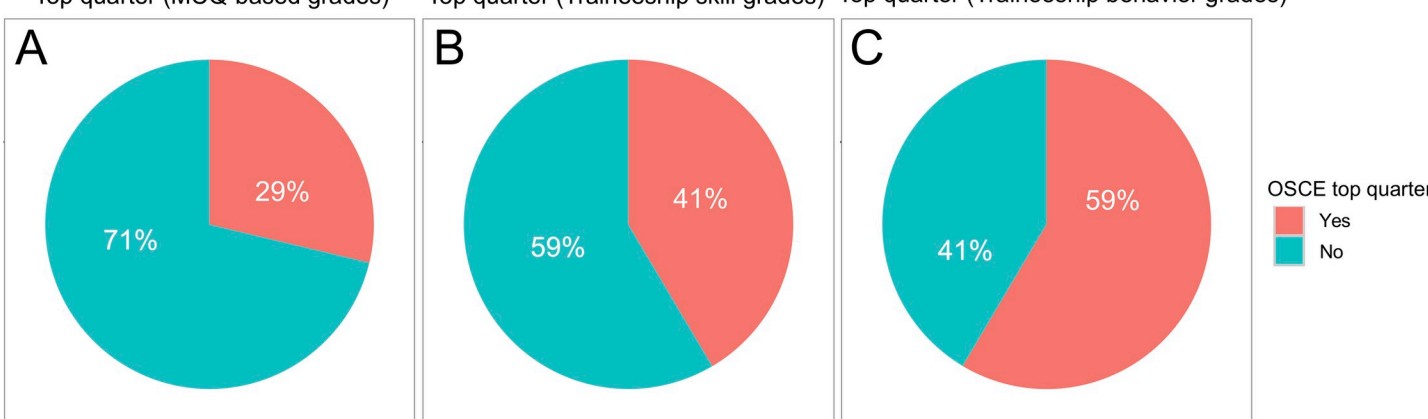

**Fig 4. Proportion of students ranked in the top quarter based on fourth-year teaching unit grades who were ranked within the top quarter of OSCE grades (average of OSCE #1–3).** (A) Multiple-choice question (MCQ)-based examination grades (fourth-year average). (B) Traineeship skill grades (fourth-year average). (C) Traineeship behavior grades. There was a significant difference between the three proportions (*P*<0.001, Chi-square test).

dispersion of grades, assessed by the standard deviation, was higher for OSCE than for MCQ-based written examinations (*P*<0.001), as confirmed graphically in **Fig 2**.

The overall relationship between mean OSCE and MCQ-based grades is displayed in **Fig 5**. The ratio between OSCE and MCQ-based grades is higher for lower MCQ-based grades. In other words, the OSCE grades are more likely to be higher than the MCQ-based grade when the MCQ-based grade is low. The discrepancy between the two grades is higher for students with lower grades on the MCQ-based examination. The regression line shows that the ratio between OSCE and MCQ-based grades tends towards 1 for higher MCQ-based grades.

Cardiovascular and hepato-gastroenterology topics predominated in the OSCE scenarios. Since fourth-year students are divided into three groups that follow the TUs in a rotating order, the quarter when a student was taught TU1, 2, or 3 may have affected OSCE grades. To rule out this potential bias, we computed a uni- and multivariate model predicting OSCE grades using the attributed rotating group (TU1/2/3, TU2/3/1 or TU3/1/2 over the 3 quarters of the academic year) and the examination grades obtained for TU1 (cardiovascular diseases) and TU2 (hepato-gastroenterology). The grades obtained at TU1 (*P*<0.001) and TU2 (*P*<0.001), but not the quarter in which the students had received training (*P* = 0.60) influenced OSCE grades in the univariate model. We also built a multivariate model into which the

**Table 3. Correlation between averaged knowledge-oriented and behavior-oriented items composing OSCE grades and mean fourth-year grades.**

|  | **Spearman *r*** | ***P*** |
|---|---|---|
| **Competence-oriented OSCE items** |  |  |
| vs. overall MCQ-based grade | 0.15 | 0.004 |
| vs. traineeship skills grade | 0.15 | 0.003 |
| vs. traineeship behavior grade | - | 0.35 |
| **Behavior-oriented OSCE items** |  |  |
| vs. overall MCQ-based grade | - | 0.079 |
| vs. traineeship skills grade | 0.13 | 0.010 |
| vs. traineeship behavior grade | 0.11 | 0.046 |

OSCE = objective structured clinical examination; MCQ = multiple-choice question

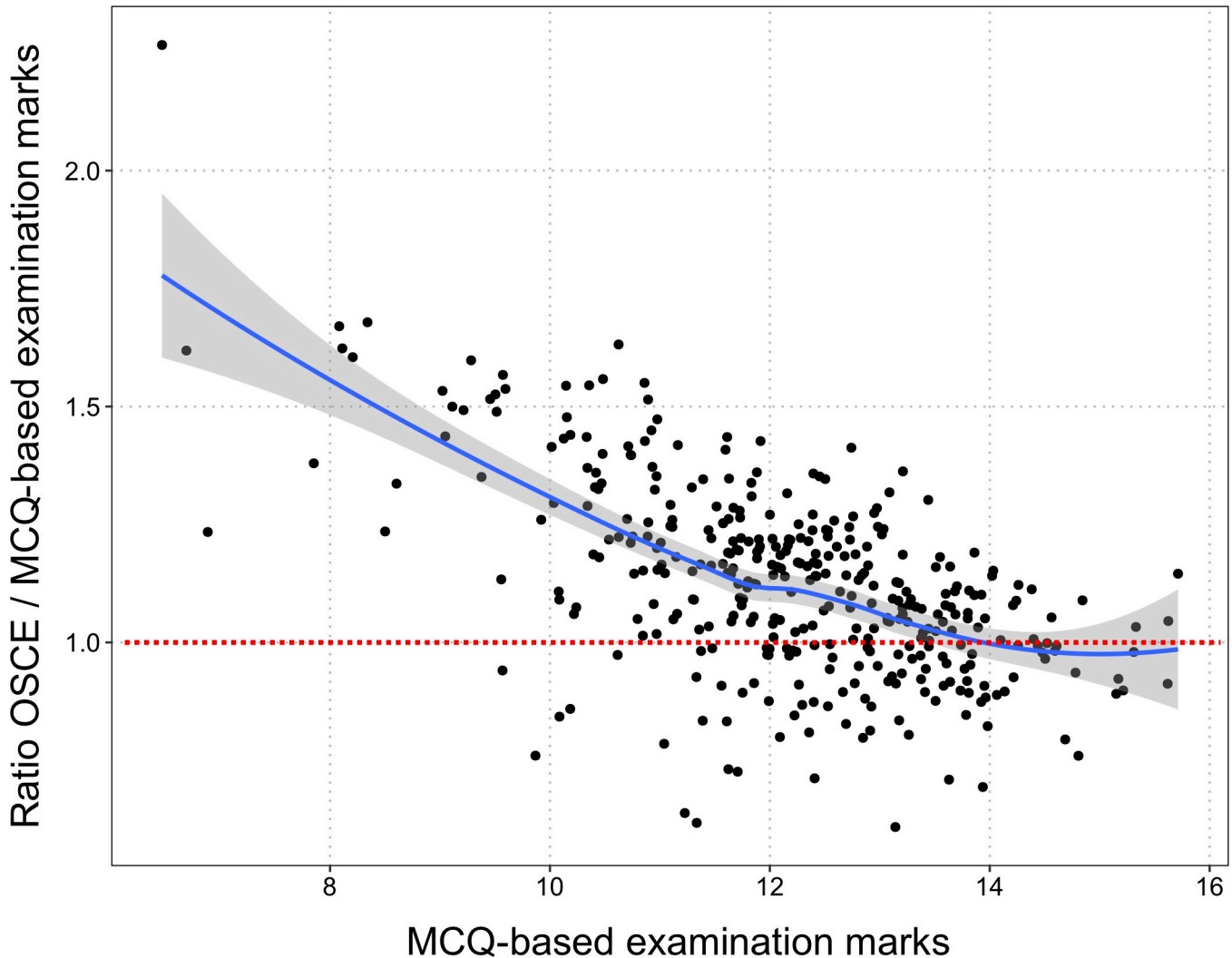

**Fig 5. Dot plot of the relationship between multiple-choice question (MCQ)-based grades obtained for teaching units and the ratio of the OSCE grades and those MCQ-based grades.** This plot highlights graphically that MCQs and OSCE evaluates students differently, since a non-neglectable proportion of students obtained better grades at OSCE than at MCQs, and more so among students with middle- or low-range grades at MCQs. To facilitate the reading, the dotted red line indicates the OSCE/MCQ ratio equal to 1.0. Students with an OSCE/MCQ ratio lower than 1.0 have a lower grade on the OSCE than the MCQ-based exam.

'training quarter' parameter was forced and found that the only contributing parameter was the TU1 grade ($P<0.001$) (multivariate model: $R^2 = 0.040$, $P<0.001$), reflecting the predominant proportion of cardiovascular topics in the OSCE.

### Impact of integrating OSCE grades into the current evaluation system

We simulated the impact on the rank of students of integrating an incremental coefficient of 10%, 20% and 40% of OSCE grades into the fourth-year average grade (**Fig 6**). As the coefficient of OSCE grades increased, an increasing proportion of the 379 students had a ranking variation by ±50 ranks (n = 2, n = 50 and n = 131 of 379 students, respectively; $P<0.001$, Chi-square test), as displayed on **Fig 6**.

Moreover, for all coefficients, the rank-variation was more important for students in the mid-50% of ranking, compared to students in the top or the bottom 25%, as evidenced visually

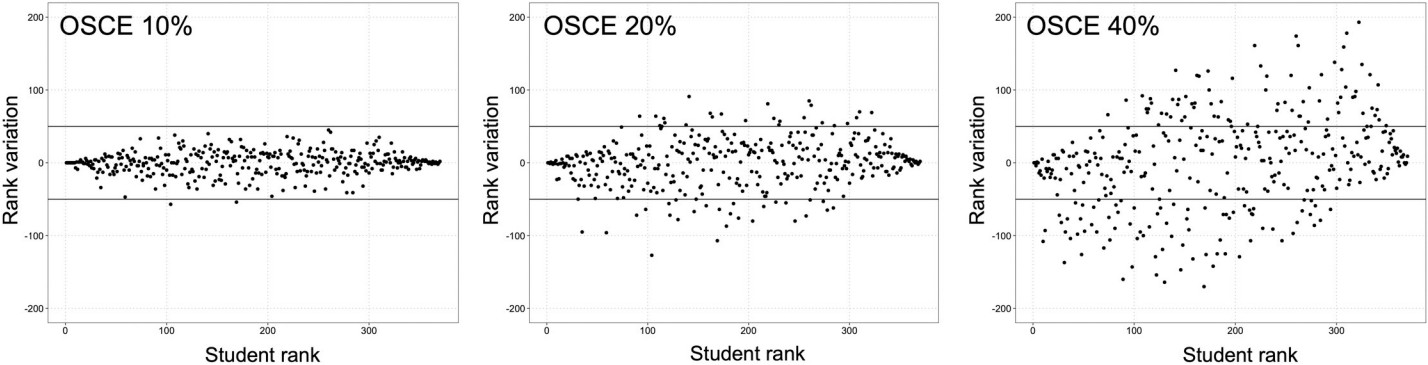

**Fig 6. Variation in ranking based on the mean fourth-year multiple-choice question (MCQ)-based grades, with incremental percentages of OSCE grade integrated into the final grade.** The upper and lower solid black lines represent thresholds for +50 or -50 rank variation, respectively. Results are displayed for integration of OSCE grade with a 10%, 20% and 40% coefficient.

on **Fig 6**. The magnitude of this effect was progressive as the OSCE coefficient increased. When integrating OSCE grades with a 10% coefficient, no student in the top or bottom 25%, but 2 students in the mid-50% of ranking changed their ranking by ±50, ($P$ = 0.50, Fisher's exact test). When integrating OSCE grades with a 20% coefficient, 7 students in the top or bottom 25%, compared to 46 students in the mid-50% of ranking changed their ranking by ±50, ($P<0.001$, Fisher's exact test). When integrating OSCE grades with a 40% coefficient, 51 students in the top or bottom 25%, compared to 80 students in the mid-50% of ranking changed their ranking by ±50, ($P$ = 0.02, Fisher's exact test).

Regarding the effect of OSCE on the highest-ranking students, integrating the OSCE grade at 10%, 20%, or 40% of the final grade changed the composition of the top 25% of the class (95 students) by 7% (n = 7/95 students), 15% (n = 14/95 students), and 40% (n = 38/95 students) ($P<0.001$, Chi-square test).

## Discussion

This study evaluating the impact of a large-scale OSCE on students' assessment in a French medical school (i) highlighted weak but statistically significant correlations between OSCE and MCQ grades, traineeship skills or traineeship behavior assessment, mainly influenced by the design of the OSCE scenario; (ii) showed a wider dispersion of grades obtained at the OSCE compared to conventional evaluation modalities; and (iii) demonstrated that integrating OSCE marks in the current grading system modified the ranking of students and affected predominantly those in the middle of the ranking.

Previous experiences of OSCE were reported by several academic institutions worldwide. This OSCE study is among the largest described, with 379 participating students. Major studies from several countries that have assessed the correlation of OSCE with other academic evaluation modalities are summarized in **Table 4**. It is widely accepted that OSCEs offer the possibility to evaluate different levels and areas of clinical skills [17, 18]. In contrast to conventional MCQs or viva voce examinations, OSCEs are designed to assess student competences and skills rather than sheer knowledge [19], as exemplified throughout the studies listed in **Table 4**. Yet, there is no precise border between clinical skills and knowledge in a clinical context [16, 18]. The categorization of OSCE items into broad evaluation fields may help extract valuable and quantitative parameters reflecting each student's clinical and behavioral skills, as performed in the present study. The three OSCE stations composing this large-scale test were designed to specifically assess clinical competence and relational skills (referred to as

**Table 4. Previous studies from the literature investigating correlations between OSCE and other academic assessment methods.**

| Reference | Country | Students, No. | Academic assessment method compared to OSCE | Statistical evaluation of the inter-relationship | Conclusions |
|---|---|---|---|---|---|
| Smith et al (1984) | United Kingdom | 229 | Viva voce examination, in-case assessment (clinical aptitude and written project), MCQ examination, comparable traditional clinical examinations. | Significant correlation between OSCE and marks from MCQ ($r = 0.34$, $P<0.001$), comparable clinical examination ($r = 0.26$, $P<0.001$), and previous in-case assessment ($r = 0.24$, $P<0.001$). | In contrast to viva-voce examination, OSCE results correlated well with an overall assessment of the student's ability. |
| | | | | No correlation between OSCE and viva voce examination ($r = 0.08$, $P>0.05$). | The clinical component of OSCE did not correlate well with MCQ. |
| Probert et al (2003) | United Kingdom | 30 | Long and short case-based viva voce examinations. | Overall performance at traditional finals was correlated with the total OSCE mark ($r = 0.5$, $P = 0.005$). Dichotomizing traditional examinations into surgery and medicine assessment resulted in significant correlations between OSCE and surgery marks ($r = 0.71$, $P<0.001$) but not between OSCE and medicine marks ($r = 0.15$, $P = 0.4$). | This was a pilot study for OSCE implementation, and the analyzed sample of students who performed both examination methods was representative of the whole population. |
| | | | The authors added independent consultant evaluations to assess clinical performance by students. | | OSCE assesses different clinical domains than do traditional finals and improved the prediction of future clinical performance. |
| Dennehy et al (2008) | USA | 62 | National Board Dental Examination (NBDE, computerized assessments of theoretical knowledge in part I, and clinical knowledge in part II), and MCQ examinations. | NBDE score was statistically associated with OSCE score ($P$ ranging from $<0.001$ to $0.04$). | Didactic predictors (NBDE, MCQ examinations) explained around 20% of the variability in OSCE scores. |
| | | | | There was no significant association between OSCE and MCQ scores. | OSCE may be a tool that allows educators to assess student capabilities that are not evaluated in typical standardized written examinations. |
| | | | | In multiple regression models none of the didactic predictors were significantly associated with overall OSCE performance. | |
| Sandoval et al (2010) | Chile | 697 | Written examination and daily clinical practice observation guidelines. | Positive correlation between percentages of success for all three evaluation methods with OSCE ($P<0.001$). | Pearson's correlation co-efficient was higher between assessment methods after seven years of OSCE implementation. |
| | | | | | These evaluations are complementary. |
| Kirton et al (2011) | United Kingdom | 39 per year during a 3-year long evaluation | Medicine and pharmacy practice (MPP) written examination combining MCQ and essays expected to relate to clinical practice. | Moderate positive correlation between OSCE and MPP ($r = 0.6$, $P<0.001$). | For 20% of students, experience in OSCE did not increase marks or individual performance. |
| | | | | | These two examinations assess different areas of expertise according to Miller's Pyramid of Competence and both should be performed. |
| Kamarudin et al (2012) | Malaysia | 152 | Student's clinical competence component evaluated during the final professional long-case examination. | Positive correlation between OSCE and long case for the diagnostic ability ($r = 0.165$, $P = 0.042$) and total exam score ($r = 0.168$, $P = 0.039$). | There is a weak correlation between OSCE and long-case evaluation format. These two assessment methods test different clinical competences. |
| Tijani et al (2017) | Nigeria | 612 | Long-case examination (end of posting in the 4th and 6th years of medical school), final MCQ, and total written papers (TWP): sum of MCQ examinations and essays. | Positive correlation between OSCE and MCQ ($r = 0.408$), TWP ($r = 0.523$), and long case ($r = 0.374$), $P<0.001$. | The total clinical score combining OSCE and long-case marks was a better predictor of student clinical performance than each assessment method analyzed separately. |
| | | | | | These two evaluations could be complementary. |
| | | | | | Previous experience with OSCE was not taken into account in the analysis. |

(*Continued*)

**Table 4.** (Continued)

| Reference | Country | Students, No. | Academic assessment method compared to OSCE | Statistical evaluation of the inter-relationship | Conclusions |
|-----------|---------|---------------|---------------------------------------------|--------------------------------------------------|-------------|
| Lacy et al (2019) | Mexico | 83 | Communication skills evaluated during direct observation of a clinical encounter (DOCE) using the New Mexico Clinical Communication Scale. | Students' matched scores on OSCE and DOCE were not correlated. | The discordance between OSCE and DOCE suggests that OSCE may not be an optimal method to identify students requiring additional communication training. |
| | | | | Mean scores were not statistically different between faculty evaluators for individual communication skills ($P = 0.2$). | |

No. = number; OSCE = objective structured clinical examination; MCQ = multiple-choice question; NBD = national board dental examination; MPP = medicine and pharmacy practice; TWP = total written papers; DOCE = direct observation of a clinical encounter.

"behavior") in different proportions. Interestingly, we observed different correlation profiles between OSCE grades at each station, and MCQs, traineeship skills and traineeship behavior. The more competence-oriented OSCE #1 station correlated only with MCQ grades, while the balanced OSCE#2 correlated near-significantly with MCQ grades, and significantly with traineeship skill grades, and finally the behavior-oriented OSCE #3 correlated with both traineeship skill and behavior grades. These differential profiles confirm the paramount importance of OSCE station design, according to its specific pedagogic objectives, as recently pointed out by Daniels and Pugh who proposed guidelines for OSCE conception [20]. Remarkably, similar correlations have been previously observed in studies summarized in **Table 4** [19, 21–23], which supports the reliability of OSCE as an evaluation tool for medical students [24]. To note, the weak level of correlations observed between OSCE grades and the other evaluation modalities in the present study is consistent with the weakness of correlations reported in the literature (see **Table 4**). It may reflect the fact that OSCEs evaluate skills in a specific manner depending on their design, as compared to conventional assessment methods [19, 22]. Overall, the correlations observed between OSCE grades and classical assessment modalities, and the consistence of weak correlation levels with those reported in the literature, strongly support the notion that these correlations do not result from chance or from a fluctuation of grades.

Importantly, we observed a significantly larger distribution of grades obtained at OSCE compared to grades from current academic evaluation modalities, relying essentially on written MCQ tests. This underlies the potential discriminating power of OSCE for student ranking, of importance in the French medical education system and many other countries, where admissions into residency programs depend on a single national ranking. Currently, more than 8,000 6th-year medical students take the French national classifying exam each year. Its outcome has been subject to criticism over the hurdles to accurately rank such large number of students based on MCQs only [25].

Finally, this study underlines the potential impact of OSCE on student ranking. OSCE have not been employed in other national settings for the purpose of student ranking, a specificity of the French medical education system, but rather as a tool to improve or to evaluate clinical competence. Using a simulation strategy, we observed that the impact of integrating OSCE grade with a 10-to-40% coefficient was greater for students with intermediate ranks, which is of importance since it suggests that OSCE may contribute to increase the discriminatory power of the French classifying national exam. This observation is the consequence of the two above-mentioned results, showing a weak correlation between OSCE and MCQs grades and a larger distribution of grades obtained at OSCE compared to grades from current academic evaluation modalities. At both ends of the distribution of MCQ grades there were fewer students, resulting in a higher MCQ grade difference between top- or bottom-ranked students

than among middle-ranked students. Therefore, integrating the OSCE grade with a coefficient up to 40% did not change the composition of the top and bottom ranks. It should be noted, however, that the discriminating ability of OSCE is debated. As pointed by Konje et al, OSCE are complementary to other components of medical students' examination, such as clinical traineeships, but may not be sufficient to assess all aspects of their clinical competences in order to classify them [26]. Moreover, Daniels et al have demonstrated that the selection of checklist items in the design of OSCE stations has a strong effect on the station reliability to assess clinical competence, and, therefore on its discriminative power [24]. Currently, the French national classifying exam, based on MCQs only, is appropriate to discriminate higher and lower-level students, but several concerns have been raised over its ability to efficiently discriminate students in the middle of the ranking where grades are very tight [25, 27]. Moreover, these MCQs assess mainly medical knowledge and have little ability to assess clinical skills [28]. Whether OSCE are well correlated to real-life medical and behavior skills could not be assessed in our study but OSCE have already proven their superiority to evaluate knowledge, skill, and behavior compared to written examinations [19–22]. In addition, the French academic context requires this novel examination modality to possess a high discriminatory power, in order to contribute to the national student ranking. Overall, these previous results indicate that OSCE is potentially a relevant and complementary tool for student training and ranking [29, 30].

This study has several limitations. It reports the first experience using OSCE over an entire medical school class of the Université de Paris. Therefore, students had not been previously trained for this specific evaluation modality. In future, the impact of OSCE grade integration may be modified when French students will have trained specifically before taking the final OSCE. Moreover, standardized patients were voluntary teachers from the institution. According to the standards of best practice from the Association of Standardized Patient Educators (ASPE), standardized patients do not have to be professional actors [31]. However, the fact that they were medical teachers may have induced an additional stress in students, possibly altering their performance. In addition, contrary to the ASPE guidelines [31], no screening process was applied to medical educators who were recruited on a voluntary basis from all clinical departments in our University Hospitals, because 162 educators were required to run all OSCE stations simultaneously. To minimize these biases and homogenize their roles, a training program for teachers who acted as standardized patients was well-defined and mandatory. An additional bias may result from inter- or intra-standardized patient variability that may be noted in performances over time. We attempted to limit this bias by homogenizing the training of standardized patients during several pre-OSCE meetings, by sharing videos of the expected standard roles, and by controlling their performance by observers from the OSCE committee during the examination. The proportion of evaluators from the same specialty as the one evaluated in each OSCE station was <10%, which can be deemed sufficiently low not to bias the evaluation. For future OSCE sessions, the organizing committee from our University should exclude specialists from OSCE stations of their own field. To reduce evaluation bias, care should also be taken to minimize the risk for an evaluator to have already evaluated during an hospital traineeship one of the students taking his/her OSCE station. Moreover, for practical reasons during this first large-scale session, students were assessed in only three OSCE stations, whereas at least eight stations are usually used for medical school examinations [20, 24]. The ranking of the fourth-year medical students according to the mean of all the MCQs of the three TUs probably will not be the rankings these students will receive in two years at the final national classifying exam. Finally, since teaching programs differ between countries, results from this French study may not be relevant to other education systems.

These results consolidate the current project of expanding the use of OSCE in French medical schools and suggest further developments. Besides increasing the number of stations and diversifying scenarios to cover multiple components of clinical competence, future studies should explore the potential use of OSCE not only as evaluation tool, but also as learning tool, as compared to traditional bedside training. Among other parameters, the impact of OSCE on student grades within a given teaching unit should be investigated. Feedback from students, medical teachers, and simulated patients have been collected and are under analysis to fine-tune the conception and organization of OSCE in France, both at local and national levels.

In conclusion, this large-scale French experiment showed that OSCE assess clinical competence and behavioral skills in a complementary manner, compared to conventional assessment methods, as highlighted by the weak correlation observed between OSCE grades and MCQ grades, traineeship skills or behavior assessment. It also demonstrated that OSCE have an interesting discriminatory capacity, as highlighted by the larger distribution of grades obtained at OSCE compared to grades from current academic evaluation modalities. Finally, it evidenced the impact of integrating OSCE grades into the current evaluation system on student ranking.

## Supporting information

**S1 Table. Grades obtained at each OSCE station.**
(TXT)

**S1 Data. OSCE #1 script and evaluation grid.**
(DOCX)

**S2 Data. OSCE #2 script and evaluation grid.**
(DOCX)

**S3 Data. OSCE #3 script and evaluation grid.**
(DOCX)

## Acknowledgments

The authors thank Mrs. Bintou Fadiga, European Georges-Pompidou Hospital, Necker-Enfants Malades Hospital and Cochin Hospital, AP-HP, Paris, for technical assistance.

## Author Contributions

**Conceptualization:** Alexandre Matet, Ludovic Fournel, Jean-Benoit Arlet, Stéphanie Baron, Anne-Sophie Bats, Albert Faye, Natacha Kadlub, Gérard Friedlander, Philippe Ruszniewski, Cécile Badoual, Brigitte Ranque, Mehdi Oualha, Marie Courbebaisse.

**Data curation:** Alexandre Matet, Ludovic Fournel, Laurence Amar, Jean-Benoit Arlet, Anne-Sophie Bats, Mehdi Oualha.

**Formal analysis:** Alexandre Matet, Ludovic Fournel, François Gaillard, Brigitte Ranque, Mehdi Oualha, Marie Courbebaisse.

**Investigation:** Alexandre Matet, Ludovic Fournel, Stéphanie Baron, Anne-Sophie Bats, Celine Buffel du Vaure, Caroline Charlier, Victoire De Lastours, Albert Faye, Natacha Kadlub, Julien Leguen, David Lebeaux, Alexandre Malmartel, Tristan Mirault, Benjamin Planquette, Alexis Régent, Jean-Laurent Thebault, Alexy Tran Dinh, Alexandre Nuzzo, Guillaume Turc, Gérard Friedlander, Philippe Ruszniewski, Cécile Badoual, Brigitte Ranque, Mehdi Oualha, Marie Courbebaisse.

**Methodology:** Alexandre Matet, Ludovic Fournel, François Gaillard, Laurence Amar, Eve Jablon, David Lebeaux, Brigitte Ranque, Mehdi Oualha, Marie Courbebaisse.

**Project administration:** Alexandre Matet, Brigitte Ranque, Marie Courbebaisse.

**Resources:** Brigitte Ranque.

**Supervision:** Mehdi Oualha, Marie Courbebaisse.

**Validation:** Alexandre Matet, Mehdi Oualha, Marie Courbebaisse.

**Writing – original draft:** Alexandre Matet, Ludovic Fournel, François Gaillard, Mehdi Oualha, Marie Courbebaisse.

**Writing – review & editing:** Alexandre Matet, Laurence Amar, Jean-Benoit Arlet, Stéphanie Baron, Anne-Sophie Bats, Celine Buffel du Vaure, Caroline Charlier, Victoire De Lastours, Albert Faye, Eve Jablon, Natacha Kadlub, Julien Leguen, David Lebeaux, Alexandre Malmartel, Tristan Mirault, Benjamin Planquette, Alexis Régent, Jean-Laurent Thebault, Alexy Tran Dinh, Alexandre Nuzzo, Guillaume Turc, Gérard Friedlander, Philippe Ruszniewski, Cécile Badoual, Brigitte Ranque, Mehdi Oualha, Marie Courbebaisse.

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
