## [Decision Letter · Decision Letter 0]

12 Nov 2020

PONE-D-20-30792

Impact of Integrating Objective Structured Clinical Examination into Academic Student Assessment: Large-Scale Experience in a French Medical School

PLOS ONE

Dear Dr. MATET,

Thank you for submitting your manuscript to PLOS ONE. After careful consideration, we feel that it has merit but does not fully meet PLOS ONE’s publication criteria as it currently stands. Therefore, we invite you to submit a revised version of the manuscript that addresses the points raised during the review process.

We look forward to receiving your revised manuscript.

Kind regards,

Etsuro Ito

Academic Editor

PLOS ONE

Journal Requirements:

Additional Editor Comments:

The comments seem very minor. Please revise your MS according to them.

Reviewers' comments:

Reviewer's Responses to Questions

**Comments to the Author**

1. Is the manuscript technically sound, and do the data support the conclusions?

Reviewer #1: Yes

Reviewer #2: Yes

Reviewer #3: Yes

2. Has the statistical analysis been performed appropriately and rigorously? 

Reviewer #1: Yes

Reviewer #2: Yes

Reviewer #3: Yes

3. Have the authors made all data underlying the findings in their manuscript fully available?

Reviewer #1: Yes

Reviewer #2: Yes

Reviewer #3: Yes

4. Is the manuscript presented in an intelligible fashion and written in standard English?

Reviewer #1: Yes

Reviewer #2: Yes

Reviewer #3: Yes

5. Review Comments to the Author

Reviewer #1: This manuscript reports on a relevant and well conducted study.

Concusions are meaningfull not only for the French system but also for other countries internationally.

I only have minor remarks, already partially adressed by the authors, but for whci I would like a bit more clarifications and highlighting in the paper.

Methods: Assessment of students using OSCE required a large number of assessors (162+27). How did the investigators ensure for the homogeneity of the training and motivation of these assessors?

Results: how do you explain different correlations of various OSCE (particularly OSCE1 and 2, because OSCE3 is clearly a different exercise) and classical assessment tool. Does it represent real differences with plausible explanation or fluctuation of the results compromising their meaning and significance?

Reviewer #2: Matet et al. report a study evaluating the correlations between OSCE and “traditional” French medical student evaluation, which combines hospital-based traineeship evaluation and academic evaluation using MCQs. This study is of importance, as long OSCE will be soon integrated in the French medical school evaluation system and will account for 40% of the final exam grade. Besides, such correlations were so far not described in this setting and give insights into the potential consequences of such a transition, which may take place in other countries.

Among the novel findings, it is reported that only one third of the top 25% MCQ graded students were among the top 25% OSCE graded students. Although some of the MCQ and OSCE grades were statistically correlated, correlations remained poor, despite an interesting sample size.

These discrepancies precisely illustrate the limits of MCQs, which are definitely less able to evaluate the clinical skills of medical students.

Finally, the authors show that OSCE integration may increase the discriminative capacity of the exam, a crucial finding for a ranking exam involving thousands of students…

The paper is overall well written and clear. Statistical analyses are appropriate. The main limitations (absence of student training / limited number of workshop / teachers instead of trained actors) are discussed in detail.

I would have some questions and minor comments:

- Were teachers involved as OSCE patients or evaluators chosen in medical specialties different to the evaluated ones? This should be mentioned as evaluators should ideally be chosen among other specialties to improve the objectivity of evaluation. Also, there could be an evaluation bias if an evaluator has already evaluated a student in the context of a Hospital-based traineeship evaluation.

- I don’t understand the point of figure 5. In the legend, it is written “Students with a ratio OSCE/MCQ lower than 1.0 (dotted red line) have a lower grade on the OSCE than the MCQ-based exam”: isn’t it obvious? Please clarify.

- I would suggest to shortly discuss the validity of traineeship skill and behavior evaluation: their results seem completely disconnected from both MCQ and OSCE grades…

Minor comments:

- Page 6, line 132: “which has two sites that have recently merged, the Paris Nord and Paris Centre sites”: I am not sure this information is relevant for this study.

- Page 6, line 133: “per class” > per year

- Page 7, line 135: “This study was conducted at the Paris Centre site”: this sentence should be in the methods section

- Page 7, line 151: “on duty call”: I don’t understand. Do you mean on night shift?

- Page 10, line 217: “all 379 participating students”: I would suggest to remove “379” as long as this is part of the results.

Reviewer #3: The study was conducted rigorously. It's an original and very interesting work, especially with the upcoming change of the French final classifying national exam. The sample size, for an OSCE examination evaluation is large, and provides interesting data. The statistical analysis are performed appropriately and the results are clearely expressed, with sufficient detail. The manuscript is written in standard and intelligible English.

I do have a few comments about the standardized patients:

It is mentionned as a limit that the standardized patients are not professional actors but volunteer teachers. In reference to ASPE SOBP's or Howard Barrows' definition, SP's do not have to be professional actors. I would rather specify as a limit the fact that they are teachers, wich could create a particular stress in students, modifiying their performance. Also, nothing is mentionned about the screening process, wich is highly recommended in SP's. Finally, nothing is said about the SP quality assurance evaluation. How many different SP's played the same patient (risk of inter-SP variability) and how many runs of each scenario did each SP do (intra-SP variability)? Differences may be noted in performances between different SP's or within the performances of a same SP over time. These are also minimal limits or biases that could be reported.

Here by you will find a few general comments about what might be typos :

page 11 line 247: the 10% coefficient is not mentionned as in th rest of the manuscript with the 20% and 40% coefficient

page 15 line 328: maybe a mistake in the rotating groups? Isn't the third group TU 3/1/2 rather than 3/2/1?

page 18 line 408: you refer to the impact of integrating OSCE grades with a 5 to 20% coefficient, but in the results, its presented as a 10 to 40% ceofficient (same on page 19 line 416, ...up to 20% isn't it up to 40%?) If it isn't a mistake, its confusing.

page 19 line 435 : typo on the word "school" written "scholl"

6. PLOS authors have the option to publish the peer review history of their article (what does this mean?). If published, this will include your full peer review and any attached files.

Reviewer #1: **Yes: **Edouard Louis

Reviewer #2: No

Reviewer #3: **Yes: **Pr Anne Bellot, Caen University Hospital, Caen Medical School, University of Caen Normandy

---

## [Author Response · Author response to Decision Letter 0]

31 Dec 2020

RESPONSE TO REVIEWERS

We thank the three Reviewers for their constructive comments. We have addressed each of them in the point-by-point response below, that refers to the changes made to the manuscript. Page and line numbers refer to the initial version of the manuscript.

REVIEWER #1: This manuscript reports on a relevant and well conducted study.

Concusions are meaningfull not only for the French system but also for other countries internationally.

I only have minor remarks, already partially adressed by the authors, but for whci I would like a bit more clarifications and highlighting in the paper.

• Methods: Assessment of students using OSCE required a large number of assessors (162+27). How did the investigators ensure for the homogeneity of the training and motivation of these assessors?

We thank Reviewer #1 for raising this issue.

- The homogeneity of the training between assessors was maximized by several means:

o Preparatory meetings throughout the academic year preceding the OSCE test, and specific training for the OSCE in small groups by one single coordinating team for each OSCE station.

o Shared video recordings of a standard patient undergoing each OSCE station.

o The homogeneity of student evaluation was further optimized by the use of shared evaluation grids (provided in Suppl Material #1-2-3).

- The homogeneity of motivation between assessors is critical, yet it is difficult - if not impossible - to ensure a complete homogeneity of motivation. To our view, the major factor warranting their motivation was that all assessors were medical doctors belonging to the University of Paris hospital network - therefore all implicated at various degrees in medical pedagogy - who participated for the first time in the large-scale pedagogical experiment of an upcoming evaluation and teaching modality. 

The following changes were made to the text:

- Methods, p 10 line 225, addition of: “The homogeneity of training between assessors was maximized by preparatory meetings throughout the academic year preceding the OSCE test, specific training for each OSCE station in small groups by one single coordinating team, diffusion of video recordings of a standard patient undergoing each OSCE station.”

- Methods, p 10 line 225, addition of: “Moreover, the homogeneity of motivation between assessors was maximized by the facts that all were medical doctors belonging to the same university hospital network, that all were implicated at various degrees in medical pedagogy, and that all participated for the first time to a large-scale pedagogical experiment of an upcoming evaluation and teaching modality.”

- We have clarified by removing the expression “27 external supervisors were enrolled”(Methods, p10 line 224), which referred to employees assigned to the organizational and not academical aspects of the OSCE exam.

• Results: how do you explain different correlations of various OSCE (particularly OSCE1 and 2, because OSCE3 is clearly a different exercise) and classical assessment tool. Does it represent real differences with plausible explanation or fluctuation of the results compromising their meaning and significance?

We believe that these discrepancies reflect real differences between the profiles of grades obtained at OSCE1, 2 and 3, and that plausible findings car explain these different profiles. It is true that correlation levels are weak, which can suggest that the dispersion of grades is high and that the observed correlations result from data fluctuation. However, several observations of the very grades, and of similar correlations from the literature, support the notion that the observed differences are real and reflect underlying factors.

First, the degrees of correlation observed are consistent with weak correlation levels reported in the literature for similar OSCE grade data.

Second, the correlation profiles observed between OSCE grades from each station and conventional modalities were consistent with the content and pedagogical orientation of each OSCE station. OSCE1 and 2 were more competence-oriented, with OSCE1 relying more on clinical knowledge (and correlating only with MCQ grades), and OSCE2 relying on the relational capacity to translate this knowledge to a patient (and correlating only with hospital traineeship skills and behavior). In contrast, OSCE3 was clearly more behavior-oriented, and correlated consistently with traineeship behavior grades only.

We made the following addition:

Discussion, p18 line 403, addition of: “Overall, the correlations observed between OSCE grades and classical assessment modalities, and the consistence of weak correlation levels with those reported in the literature, strongly support the notion that these correlations do not result from chance or from a fluctuation of grades.”

REVIEWER #2: Matet et al. report a study evaluating the correlations between OSCE and “traditional” French medical student evaluation, which combines hospital-based traineeship evaluation and academic evaluation using MCQs. This study is of importance, as long OSCE will be soon integrated in the French medical school evaluation system and will account for 40% of the final exam grade. Besides, such correlations were so far not described in this setting and give insights into the potential consequences of such a transition, which may take place in other countries. 

Among the novel findings, it is reported that only one third of the top 25% MCQ graded students were among the top 25% OSCE graded students. Although some of the MCQ and OSCE grades were statistically correlated, correlations remained poor, despite an interesting sample size. 

These discrepancies precisely illustrate the limits of MCQs, which are definitely less able to evaluate the clinical skills of medical students. 

Finally, the authors show that OSCE integration may increase the discriminative capacity of the exam, a crucial finding for a ranking exam involving thousands of students... 

The paper is overall well written and clear. Statistical analyses are appropriate. The main limitations (absence of student training / limited number of workshop / teachers instead of trained actors) are discussed in detail. 

I would have some questions and minor comments: 

• Were teachers involved as OSCE patients or evaluators chosen in medical specialties different to the evaluated ones? This should be mentioned as evaluators should ideally be chosen among other specialties to improve the objectivity of evaluation. Also, there could be an evaluation bias if an evaluator has already evaluated a student in the context of a Hospital-based traineeship evaluation.

- We thank the Reviewer for raising a critical issue in OSCE preparation and practical organization. The selection of evaluators was random among university hospital physicians in clinical departments affiliated to our institution. Based on the Reviewer’s suggestion, we retrospectively assessed the proportion of teachers from the same specialties as the OSCE stations they were assigned to (pneumologists in OSCE1, cardiologists/vascular specialists in OSCE2, and gastroenterologists/digestive surgeons in OSCE3). This proportion was 9.5% across the 3 stations (7%, 12% and 9% for OSCE 1, 2 and 3, respectively), which can be deemed sufficiently low not to bias the evaluation. For future OSCE sessions, the organizing committee from our University should exclude specialists from OSCE stations of their own field.

- We are also grateful for the second suggestion, which is extremely relevant. The risk for an evaluator to have already met one of the students evaluated in his/her OSCE station was very low since students are affected to more than a hundred departments for traineeship positions. Yet it remained possible, and this potential bias should therefore be anticipated in the next OSCE examinations.

We made the following additions to the text:

- Methods, p11 line 230: “The proportion of evaluators from the same specialty as the one evaluated in each OSCE station (pneumologists in OSCE1, cardiologists/vascular specialists in OSCE2, and gastroenterologists/digestive surgeons in OSCE3) was ~9.5% across the 3 stations. This proportion was 7%, 12%, and 9% for OSCE 1, 2 and 3, respectively.”

- Discussion, p20 line 445, § on study Limitations: “The proportion of evaluators from the same specialty as the one evaluated in each OSCE station was <10%, which can be deemed sufficiently low not to bias the evaluation. For future OSCE sessions, the organizing committee from our University should exclude specialists from OSCE stations of their own field.” 

- Discussion, p20 line 447, § on study Limitations: To reduce evaluation bias, care should also be taken to minimize the risk for an evaluator to have already evaluated during an hospital traineeship one of the students taking his/her OSCE station.

• I don’t understand the point of figure 5. In the legend, it is written “Students with a ratio OSCE/MCQ lower than 1.0 (dotted red line) have a lower grade on the OSCE than the MCQ-based exam”: isn’t it obvious? Please clarify. 

This legend was explanatory and, indeed, the legend simply described a logical, graphically visible finding.

The main finding displayed in Figure 5 was that a non-neglectable proportion of students obtained much better grades at OSCE than at conventional examination modalities such as MCQs.

In order to stress the fact that our commentary in the legend is explanatory and does not intent to reflect the main finding of this Figure, we have clarified the Legend of Figure 5 as follows:

- “Figure 5. Dot plot of the relationship between multiple-choice question (MCQ)-based grades obtained for teaching units and the ratio of the OSCE grades and those MCQ-based grades. This plot highlights graphically that MCQs and OSCE evaluates students differently, since a non-neglectable proportion of students obtained better grades at OSCE than at MCQs, and more so among students with middle- or low-range grades at MCQs.

To facilitate the reading, the dotted red line indicates the OSCE/MCQ ratio equal to 1.0. Students with an OSCE/MCQ ratio lower than 1.0 have a lower grade on the OSCE than the MCQ-based exam.”

Minor comments:

- Page 6, line 132: “which has two sites that have recently merged, the Paris Nord and Paris Centre sites”: I am not sure this information is relevant for this study.

The ‘Paris Nord’ site corresponds to the former Paris Diderot University, and the ‘Paris Centre site’ corresponds to the former Paris Descartes University. Both entities have merged in 2019, the same year the first large-scale OSCE described in the manuscript was performed. We agree that this information is not essential to the study. However, this study was conducted exclusively on grades from the Paris Centre site, and this is specified in the manuscript (see Comment below on p7, line 151). Another study on this OSCE, that focused on student communication skill training, has analyzed all grades from both the Paris Nord and the Paris Centre site (PMID: 32886733). Therefore, we wish to maintain this clarification.

- Page 6, line 133: “per class” > per year.

The term class has been replaced by “study year”.

- Page 7, line 135: “This study was conducted at the Paris Centre site”: this sentence should be in the methods section 

We have removed this mention since the “Paris Centre site” is already specified in the “Study Population” paragraph of the Methods section.

- Page 7, line 151: “on duty call”: I don’t understand. Do you mean on night shift?

Yes, we have replaced “duty call” by “night shift”.

- Page 10, line 217: “all 379 participating students”: I would suggest to remove “379” as long as this is part of the results.

Yes, we have simply rephrased as “all participating students”.

REVIEWER #3: The study was conducted rigorously. It's an original and very interesting work, especially with the upcoming change of the French final classifying national exam. The sample size, for an OSCE examination evaluation is large, and provides interesting data. The statistical analysis are performed appropriately and the results are clearly expressed, with sufficient detail. The manuscript is written in standard and intelligible English. 

I do have a few comments about the standardized patients:

• It is mentioned as a limit that the standardized patients are not professional actors but volunteer teachers. In reference to ASPE SOBP's or Howard Barrows' definition, SP's do not have to be professional actors. I would rather specify as a limit the fact that they are teachers, wich could create a particular stress in students, modifiying their performance.

We are grateful to the Reviewer for pointing out the exact definition of standard patient according to ASPE SPBP’s standards. We fully agree and have modified the manuscript accordingly, as follows.

- Discussion, p20 line 450, removal of: “not professional actors, but…”

- Discussion, p20 line 450, addition of: According to the standards of best practice from the Association of Standardized Patient Educators, standardized patients do not have to be professional actors [31]. However, the fact that they were medical teachers may have induced an additional stress in students, possibly altering their performance. (…) To minimize these biases and homogenize their roles, a training program for teachers who acted as standardized patients was well-defined and mandatory.

- Addition of a Reference: Standards of best practice from the Association of Standardized Patient Educators (New reference 31).

• Also, nothing is mentionned about the screening process, wich is highly recommended in SP's.

Again, we thank the Reviewer for raising a critical issue regarding the preparation of medical educators contributing to an OSCE. The recruitment of medical educators was made on a voluntary basis: each clinical department from our University Hospitals was contacted to indicate the name of one or several voluntary physician. However, due to the large number of students taking the OSCE simultaneously, and the fact that two educators per OSCE station were required (one actor and one evaluator), no further screening was made to refine the selection of educators.

We have indicated this notion in the Limitation paragraph of the manuscript:

- Discussion, p20 line 450, addition of: “In addition, contrary to the ASPE guidelines, no screening process was applied to medical educators who were recruited on a voluntary basis from all clinical departments in our University Hospitals, because 162 educators were required to run all OSCE stations simultaneously.”

• Finally, nothing is said about the SP quality assurance evaluation. How many different SP's played the same patient (risk of inter-SP variability) and how many runs of each scenario did each SP do (intra-SP variability)? Differences may be noted in performances between different SP's or within the performances of a same SP over time. These are also minimal limits or biases that could be reported. 

Standardized patient quality assurance is indeed crucial to assess the robustness of an OSCE session. Regarding the OSCE analyzed in the manuscript, 27 different standardized patients played the same patient role for each OSCE scenario. Each scenario was run between 12 to 15 times by each standardized patient. OSCE coordinators attended as observers at least one OSCE scenario run by each standardized patient. It is not possible to assess accurately the quality of each OSCE actor but efforts were made to homogenize the acting performance between standardized patients, by implementing a standardized training, as reported above for assessors of OSCE (changes to the manuscript detailed in our Response to Point 1 from Reviewer #1 above: several training session throughout the academic year preceding the OSCE, videos shared before the OSCE displaying the expected standard performance).

We made the following changes to the manuscript:

- Methods, p10 line 220, addition of: “To assess quality and inter-standardized patient reproducibility, OSCE coordinators attended as observers at least one OSCE scenario run by each standardized patient.”

- Results, p12 line 260, addition of: “Twenty-seven different standardized patients played the same patient role for each OSCE scenario. Each scenario was run between 12 to 15 times by each standardized patient.”

- Discussion, p20 line 450, addition of: “An additional bias may result from inter- or intra-standardized patient variability that may be noted in performances over time. We attempted to limit this bias by homogenizing the training of standardized patients during several pre-OSCE meetings, by sharing videos of the expected standard roles, and by controlling their performance by observers from the OSCE committee during the examination.”

Here by you will find a few general comments about what might be typos :

• page 11 line 247: the 10% coefficient is not mentionned as in the rest of the manuscript with the 20% and 40% coefficient

Indeed, these are typos, we are grateful to the Reviewer for pointing them out. We have rephrased as “with 10%, 20% and 40% coefficients”

• page 15 line 328: maybe a mistake in the rotating groups? Isn't the third group TU 3/1/2 rather than 3/2/1?

Yes, we have rephrased as “TU 3/1/2”.

• page 18 line 408: you refer to the impact of integrating OSCE grades with a 5 to 20% coefficient, but in the results, its presented as a 10 to 40% ceofficient (same on page 19 line 416, ...up to 20% isn't it up to 40%?) If it isn't a mistake, its confusing.

Yes, we apologize and thank the Reviewer for the understanding. These typos are remnants from the construction of the manuscript where we had considered initially different coefficients

- p18 line 408, corrected as “with a 10-to-40% coefficient”

- p19 line 416: corrected as “integrating the OSCE grade with a coefficient up to 40%”

• page 19 line 435 : typo on the word "school" written "scholl" 

Thank you, this typo was corrected as “medical school“.

---

## [Editor Report · Decision Letter 1]

2 Jan 2021

Impact of Integrating Objective Structured Clinical Examination into Academic Student Assessment: Large-Scale Experience in a French Medical School

PONE-D-20-30792R1

Dear Dr. MATET,

We’re pleased to inform you that your manuscript has been judged scientifically suitable for publication and will be formally accepted for publication once it meets all outstanding technical requirements.

Kind regards,

Etsuro Ito

Academic Editor

PLOS ONE

---

## [Editor Report · Acceptance letter]

6 Jan 2021

PONE-D-20-30792R1 

Impact of Integrating Objective Structured Clinical Examination into Academic Student Assessment: Large-Scale Experience in a French Medical School 

Dear Dr. MATET:

I'm pleased to inform you that your manuscript has been deemed suitable for publication in PLOS ONE. Congratulations! Your manuscript is now with our production department. 

Kind regards, 

on behalf of

Prof. Etsuro Ito 

Academic Editor

PLOS ONE